# Cost-Efficient Detection of *NTRK1/2/3* Gene Fusions: Single-Center Analysis of 8075 Tumor Samples

**DOI:** 10.3390/ijms241814203

**Published:** 2023-09-17

**Authors:** Aleksandr A. Romanko, Rimma S. Mulkidjan, Vladislav I. Tiurin, Evgeniya S. Saitova, Elena V. Preobrazhenskaya, Elena A. Krivosheyeva, Natalia V. Mitiushkina, Anna D. Shestakova, Evgeniya V. Belogubova, Alexandr O. Ivantsov, Aglaya G. Iyevleva, Evgeny N. Imyanitov

**Affiliations:** 1Department of Tumor Growth Biology, N.N. Petrov Institute of Oncology, 197758 St.-Petersburg, Russiatyurinvladislav@gmail.com (V.I.T.);; 2Department of Medical Genetics, St.-Petersburg Pediatric Medical University, 194100 St.-Petersburg, Russia

**Keywords:** *NTRK1*, *NTRK2*, *NTRK3*, fusion, unbalanced expression, PCR, lung cancer, MSI-H CRC

## Abstract

The majority of *NTRK1*, *NTRK2*, and *NTRK3* rearrangements result in increased expression of the kinase portion of the involved gene due to its fusion to an actively transcribed gene partner. Consequently, the analysis of 5′/3′-end expression imbalances is potentially capable of detecting the entire spectrum of *NTRK* gene fusions. Archival tumor specimens obtained from 8075 patients were subjected to manual dissection of tumor cells, DNA/RNA isolation, and cDNA synthesis. The 5′/3′-end expression imbalances in *NTRK* genes were analyzed by real-time PCR. Further identification of gene rearrangements was performed by variant-specific PCR for 44 common *NTRK* fusions, and, whenever necessary, by RNA-based next-generation sequencing (NGS). cDNA of sufficient quality was obtained in 7424/8075 (91.9%) tumors. *NTRK* rearrangements were detected in 7/6436 (0.1%) lung carcinomas, 11/137 (8.0%) pediatric tumors, and 13/851 (1.5%) adult non-lung malignancies. The highest incidence of *NTRK* translocations was observed in pediatric sarcomas (7/39, 17.9%). Increased frequency of *NTRK* fusions was seen in microsatellite-unstable colorectal tumors (6/48, 12.5%), salivary gland carcinomas (5/93, 5.4%), and sarcomas (7/143, 4.9%). None of the 1293 lung carcinomas with driver alterations in *EGFR/ALK/ROS1/RET/MET* oncogenes had *NTRK* 5′/3′-end expression imbalances. Variant-specific PCR was performed for 744 tumors with a normal 5′/3′-end expression ratio: there were no rearrangements in 172 *EGFR/ALK/ROS1/RET/MET*-negative lung cancers and 125 pediatric tumors, while *NTRK3* fusions were detected in 2/447 (0.5%) non-lung adult malignancies. In conclusion, this study describes a diagnostic pipeline that can be used as a cost-efficient alternative to conventional methods of *NTRK1–3* analysis.

## 1. Introduction

The neurotrophic tyrosine receptor kinase (*NTRK*) genes can be activated by gene rearrangements and play a driving role in pathogenesis of some human tumors. Two drugs, entrectinib and larotrectinib, have been recently approved for the treatment of *NTRK*-associated tumors, therefore the detection of these fusions is of high medical value [1,2]. Clinical detection of *NTRK* translocations is somewhat more complicated as compared to other well-known gene rearrangements, e.g., *ALK*, *ROS1*, and *RET* fusions. The *NTRK* family includes three closely related genes (*NTRK1*, *NTRK2*, and *NTRK3*) that encode for TrkA, TrkB, and TrkC kinases, respectively. Although some tumor types demonstrate preferences towards activation of one of these receptors, comprehensive analysis of all mentioned *NTRKs* is usually required for proper patient management. *NTRK* rearrangements are highly characteristic for rare tumor varieties, e.g., infantile fibrosarcomas and breast or salivary secretory adenocarcinomas, occur at moderate frequency in pediatric malignancies, but are exceptionally rare in common cancer types [3,4,5,6]. Overall, the rare occurrence of *NTRK* translocations complicates their routine detection and raises questions about the cost efficiency of relevant laboratory procedures [7]. While the analysis of *ALK*, *ROS1*, and *RET* rearrangements is generally limited to several tumor types, *NTRK1*, *NTRK2*, and *NTRK3* are currently positioned as “agnostic” targets, calling, in theory, for systematic testing of the wide spectrum of tumors [6].

All difficulties related to the clinical detection of gene translocations are relevant for the *NTRK* status assessment. There is a multiplicity of gene partners and breakpoints involved in the emergence of *NTRK1*, *NTRK2*, and *NTRK3* fusions. *NTRK* rearrangements often but not always result in the overexpression of the kinase portion of the involved gene. However, a high level of *NTRK* expression may occur without gene translocation. Pan-NTRK immunohistochemical (IHC) analysis is positioned as an acceptable screening tool; nevertheless, many studies questioned its suitability due to a high number of false-positive results, insufficient sensitivity, and the need to account for the histological origin of the tumor [8]. FISH requires the analysis of three slides using *NTRK1*, *NTRK2*, and *NTRK3* probes, respectively. IHC and FISH are considered non-expensive in the Western world, although the cost of diagnostic kits is significant, and the analysis of the obtained images is relatively time-consuming. PCR assays are criticized because they consider only the most common *NTRK* fusion variants. Conventional DNA-based next generation sequencing (NGS) may not detect the entire spectrum of *NTRK* translocations due to difficulties in the analysis of intronic regions. RNA-based NGS is regarded as the best tool for *NTRK* testing; however, many tissue specimens fail to pass quality control, and not all diagnostic panels are capable of detecting rare and new translocation variants. NGS is expensive, poorly available in some countries, and the turn-around time for this method is usually estimated in weeks [3,5,9,10,11,12,13,14].

We have developed a fast pipeline for the detection of tyrosine kinase gene translocations, which combines low cost with a sufficient level of comprehension. Its key component is a laboratory developed test for the 5′/3′-end unbalanced expression [15]. The majority of *NTRK* fusions result in increased expression of the kinase portion of the gene when it is merged to an actively transcribed gene partner. Consequently, while the amount of neighboring exonic sequences is the same in the normal gene transcript, translocation is manifested by increased production of a kinase-domain-specific RNA message as compared to the upstream portion of the gene [16,17]. The test for the 5′/3′-end unbalanced expression, in theory, requires only one PCR reaction per gene, and it is potentially capable of detecting all translocation variants. Here we evaluated the performance of the 5′/3′-end unbalanced expression test, followed by or combined with variant-specific PCR, for the detection of conventional *NTRK* fusions. We demonstrate that this approach is potentially efficient both for pre-screening and validation of translocations, as it allows for significant reduction of the number of samples undergoing RNA-based NGS.

## 2. Results

### 2.1. The Development of the 5′/3′-End NTRK Expression Test

The *NTRK1* assay involved primers located at the junctions of exons 3–4 (5′-end) and exons 14–15 (3′-end). *NTRK2* assays included primers corresponding to the borders of exons 11–12 (5′-end) and 15–16 (3′-end). However, when we tried a similar design for the detection of *NTRK3* rearrangements and utilized primers for exons 7–8 (5′-end) and 15–16 (3′-end), we observed an unacceptably high frequency of false-positive findings, probably due to alternative splicing of this gene. In order to overcome this limitation, we placed the 5′-end primers on the regions of the most common breakpoints, i.e., the border between exons 13 and 14, and exons 14 and 15, while the 3′-end primers were located in exons 16 and 17. This design allowed us to observe the depletion of the expression of exon 13–14 or 14–15 RNA sequences in case of the presence of *NTRK3* translocation (Figure 1).

### 2.2. Detection of NTRK1/2/3 Rearrangements

The study initially included 8075 tissue specimens; 651 (8.1%) of these samples failed to pass the cDNA quality control. The analysis of 6436 non-small-cell lung carcinomas (NSCLCs) revealed 11 (0.2%) instances of 5′/3′-end *NTRK* unbalanced expression. Strikingly, none of the 1293 NSCLC cases with known activating events in *EGFR*, *ALK*, *ROS1*, *RET*, or *MET* oncogenes demonstrated evidence of *NTRK* activation. The analysis of 137 pediatric and 851 non-NSCLC malignancies revealed unbalanced expression in 12/137 (8.8%) and 18/851 (2.1%) patients, respectively. In total, 41 tumors had 5′/3′-end expression imbalances. These tumors were tested for the 44 most common *NTRK* translocations by variant-specific PCR; rearrangements were detected in 20/41 (48.8%) cases. In 11 out of the remaining 21 samples, tissue samples for additional RNA extraction necessary for targeted NGS analysis were available. NGS confirmed the presence of the *NTRK* rearrangement in nine of these tumors (Figure 2).

We further evaluated to what extent the 5′/3′-end *NTRK* unbalanced expression is prone to false-negative results. Variant-specific PCR for the 44 most common *NTRK* translocations did not reveal instances of the rearrangement in 172 *EGFR/ALK/ROS1/RET/MET*-negative NSCLCs or 125 pediatric tumors, which showed no imbalance upon 5′/3′-end expression tests. However, variant-specific analysis of 447 tumors of other categories (colorectal carcinomas, gliomas, salivary gland carcinomas, sarcomas, thyroid tumors, etc.) identified two (0.5%) instances of *NTRK3* translocation (*EML4::NTRK3* (E2;N14) in microsatellite-unstable colorectal carcinoma and *ETV6::NTRK3* (E5;N15) in salivary gland carcinoma).

The performance of the 5′/3′-end *NTRK* unbalanced expression assay depends on the chosen threshold. We utilized deltaCt > 3 as a cut-off, given that this approach produced satisfactory results for the testing of other rearranged kinases [16,17,18]. We evaluated the reliability of this threshold by comparing samples with known *NTRK* rearrangements against tumors with presumably normal *NTRK* status (Appendix A). From our tumor bank, we were able to obtain 39 samples with *NTRK1* or *NTRK3* fusions in which the presence of alteration was confirmed by variant-specific PCR or NGS. However, we could not perform the validation study for *NTRK2*, as only two samples with *NTRK2* fusions were available. The control group was composed of 50 NSCLCs, which carried alterations in *EGFR*, *ALK*, *RET*, *ROS1*, and *MET* oncogenes and, therefore, were highly unlikely to have *NTRK* rearrangements. For *NTRK1*, 10/10 (100%) *NTRK1* fusion-containing samples were correctly identified by the unbalanced *NTRK1* expression test (Appendix A). The test evaluating the depletion of the *NTRK3* ex13–14 junction detected only 2/7 (28.6%) tumors with *NTRK3* fusions involving this breakpoint (Appendix A). The test for *NTRK3* ex14–15 depletion identified 15/22 (68.2%) cases with translocations involving the breakpoint at exon 15 (Appendix A). As the chosen threshold (deltaCt > 3) performed poorly for both *NTRK3* assays, we attempted to evaluate whether the modification of the cut-off will improve the testing procedure. We performed PCR analysis for 11 individual *NTRK3* translocation variants in 225 cases showing either deltaCt > 0.9 for the *NTRK3* ex13–14 depletion assay or deltaCt > 1.5 for the *NTRK3* ex14–15 depletion assay; however, none of these samples with low-level *NTRK3* expression imbalances carried *NTRK3* translocations.

While performing the analysis of distribution of *NTRK* rearrangements in different cancer types, we considered only tumor varieties with identified translocation variants. Overall, 31 tumors with *NTRK* fusions were revealed. There were 14 *NTRK1* fusions and 15 *NTRK3* translocations, while only two tumors showed evidence of *NTRK2* activation. *NTRK* rearrangements were detected in 7/6436 (0.1%) NSCLCs, 11/137 (8.0%) pediatric tumors, and 13/851 (1.5%) adult carcinomas other than NSCLC. The frequencies and spectrum of identified rearrangements are described in detail in Table 1 and Table 2, and in Figure 3. A relatively high incidence of *NTRK* fusions was seen in microsatellite unstable colorectal carcinomas (6/48, 12.5%), sarcomas (7/143, 4.9%), and salivary gland carcinomas (5/93, 5.4%). The highest frequency of *NTRK* translocations was documented in pediatric sarcomas (7/39, 17.9%). In particular, fusions were detected in 2/3 infantile fibrosarcomas, 1/5 malignant peripheral nerve sheath tumors, 1/2 dermatofibrosarcomas protuberans, 1/1 hemangioendothelioma, and 2/8 sarcomas not otherwise specified. Instances of *NTRK* rearrangements were also observed in gliomas, thyroid carcinomas, inflammatory myofibroblastic tumors, and mesoblastic nephromas; however, the number of cases was insufficient for the evaluation of the frequency of this event.

## 3. Discussion

This report presents the results of the *NTRK* analysis in a large collection of human tumors. It shows that the test for unbalanced 5′/3′-end expression is a promising prescreening tool for this rare category of gene rearrangements. Importantly, 91.9% of the included tumors passed the RNA quality control for this test. These data are comparable with the estimates obtained in the NGS RNA studies [19,20,21,22,23]. The most common cause of RNA isolation failure is poor processing of morphological samples, which is still an issue in many hospitals [24]. It is highly likely that RNA/cDNA samples, which are not suitable for reliable PCR-based testing, will also not be compatible with RNA-based NGS analysis. This study considered biological samples arriving from a few dozen different cancer centers, therefore some variations in the tissue handling could have inflated the failure rate.

Most of the studies aimed at detecting gene rearrangements employ IHC, FISH, or NGS, while the comparison of the amount of transcripts of 5′- and 3′-portions of the genes is a less common technique [25,26,27,28,29]. An example of its successful implementation is the fully automated commercial Idylla GeneFusion assay (Biocartis, Mechelen, Belgium) [30,31]. The utilization of the 5′-3′ *NTRK* imbalance assay is complicated by the requirement for robust in-house validation using a sufficient number of positive and negative controls. Furthermore, while most of the published protocols rely on the analysis of the relative overexpression of the kinase portion of the gene [30,32,33], we found here that this methodology is not suitable for the analysis of *NTRK3* gene status. As an alternative, we suggested a novel approach: we assumed, that the junction of exons, which is affected by a breakpoint, will not be present in the rearranged transcript, and, therefore, we searched for the depletion of the expression of exon 13–14 or exon 14–15 *NTRK3*-specific sequences. This methodology is potentially applicable to translocations, which are characterized by relatively narrow clustering of breakpoints, and which do not necessarily result in an easily detectable overexpression of a portion of the gene. Although being elegant, the test for depletion of expression of exon boundaries is potentially error-prone: it may be compromised by the presence of normal RNA message originating from the remaining gene allele or non-malignant cells contaminating the tumor sample. Not surprisingly, all instances of tumors that were positive by variant-specific PCR but negative by the test for unbalanced 5′/3′-end expression, contained *NTRK3* rearrangements. We attempted to decrease the cut-off for deltaCt in order to reduce the number of false-negative results obtained upon the quantitation of the 5′- and 3′-specific *NTRK3* transcripts, however, our experiments revealed that the modification of the threshold does not improve the performance of the assay (Appendix A). While establishing a PCR-based prescreening procedure, it is, therefore, advisable to supplement 5′/3′-end expression measurement by the multiplexed variant-specific analysis of *NTRK3* translocations. Furthermore, our protocol has not been rigorously validated for the detection of *NTRK2* fusions due to the low number of “positive” samples, which is another limitation of this study.

There were two tumors, which demonstrated clear-cut 5′/3′-end *NTRK2* (NSCLC) and *NTRK3* (adult sarcoma) expression imbalances, but turned out to be *NTRK* rearrangement-negative by the TruSight RNA Fusion NGS panel. It is unclear if these tumors utilize alternative splicing for one of the *NTRK* genes, or if these observations are related to the NGS failures in identifying *NTRK* fusions. It is noteworthy that NGS testing revealed a potentially actionable *EWSR1::CCDC80* fusion in the above-described sarcoma, thus making the existence of *NTRK3* translocation unlikely.

Our data strongly indicate that the occurrence of *NTRK* fusions in NSCLCs is low, even for the tumors enriched by exclusion of other actionable mutations (7/5143, 0.14%). NSCLC is also characterized by a high diversity of *NTRK* fusion types: none of the seven identified translocations occurred more than once. Obviously, the detection of *NTRK* rearrangements in NSCLCs is challenging due to the above factors, therefore the extent of this testing may need to be adjusted to the available resources. While defining the priorities, it is highly advisable to provide comprehensive *NTRK* testing to pediatric cancer patients, especially children suffering from sarcomas. Among the adult non-NSCLC malignancies, our study has confirmed a substantial frequency of *NTRK* fusions in microsatellite-unstable colorectal carcinomas (12.5%). Overall, our data on the occurrence and spectrum of *NTRK* translocations are in good agreement with previously published studies [34,35,36,37,38].

This investigation did not involve thorough validation of *NTRK*-rearranged and *NTRK*-wild-type samples by RNA sequencing, which is a limitation of the study. While the confirmation of *NTRK* fusions detected by the combination of the 5′/3′-end expression imbalance assay and variant-specific PCR seems unnecessary, it is unclear what is the risk of missing some rare *NTRK* rearrangements by the described above pipeline. The frequency of *NTRK* fusions is low, therefore, it is challenging to ensure that any given technology does not produce false-negative results. This validation would require RNA-based NGS analysis involving huge tumor collections, and, therefore, excessive costs. It is noteworthy that the frequencies of *NTRK* rearrangements presented in this dataset match the previously reported *NTRK* prevalence estimates, thus supporting the suitability of the 5′-3′ *NTRK* imbalance assay [25,28,35,36,37,39,40]. Furthermore, we have encouraging results from the NSCLC study, which focused on patients with a very high probability of the presence of actionable gene fusion in the tumor tissue, i.e., young-onset female non-smokers. RNA-based NGS was performed for 87 tumors, which lacked activating mutations in *EGFR*, *KRAS*, *NRAS*, *BRAF*, *MET*, and *HER2* oncogenes and were negative for *ALK*, *ROS1*, *RET*, or *NTRK1/2/3* translocations by the 5′/3′-end expression test and variant-specific PCR. Strikingly, no instances of rearrangements in the above genes have been revealed by RNA sequencing [41]. While data on NSCLC appear to be conclusive, we did not specifically address the performance of the *NTRK* unbalanced expression assay separately in other tumor types. There are significant differences in the patterns of *NTRK* expression and spectrum of *NTRK* rearrangements across various categories of malignancies [3,4,5,6,8,9,10], which may influence the performance of this test. This is particularly relevant to tumor entities, which have a high baseline expression level of *NTRK* kinases [26], as the presence of the background normal transcripts is likely to compromise the performance of the 5′/3′-end expression assay. The actual rate of false-negative results produced by our methodology remains to be determined in further investigations.

PCR-based fusion detection and NGS require distinct processing of the samples. Our PCR testing procedure involved simultaneous DNA and RNA isolation followed by the cDNA synthesis using the entire volume of the sample. In order to perform NGS, we subjected samples with 5′/3′-end expression imbalances to a new round of RNA isolation, with only 11 out of 21 samples available for this procedure. This low success rate is attributed to the nature of our sample collection, where most of the tumors were not initially intended to undergo comprehensive *NTRK* testing and therefore were returned to the primary hospitals right after the completion of other diagnostic procedures. This disadvantage can be resolved by dividing the DNA/RNA preparations into two parts, with one sample undergoing PCR analysis and the remaining one subjected to NGS when necessary.

Many laboratories currently utilize pan-TRK IHC as a screening tool for the detection of *NTRK* rearrangements [9]. Several investigations evaluated the performance of IHC testing in the real-world setting. Hondelink et al. [8] evaluated the sensitivity of the IHC by analyzing the tumors with NGS-detected *NTRK* translocations. These authors considered their own dataset (24 tumors) and all relevant published studies (200 tumors in total) and revealed that IHC failed to detect fusions in 40/224 (18%) of *NTRK*-rearranged samples (*NTRK1:* 6%; *NTRK2:* 14%; *NTRK3:* 27%) [8]. These data correspond well to our experience, as we observed satisfactory performance of the 5′-/3′-end expression imbalance screening test for the *NTRK1* but not for the *NTRK3* (Appendix A). IHC often produces false-positive results. Overbeck et al. [25] reported the results of the pan-TRK staining for 973 NSCLCs. IHC failures were observed for 75/973 (8%) tumors. TRK expression was detected in 133/898 (15%) NSCLCs; 120 of these carcinomas were available for NGS analysis, with only two instances of *NTRK* fusions eventually confirmed. Zito Marino et al. [42] revealed pan-TRK IHC positivity in 16 out of 83 triple-negative breast malignancies; however, none of these samples carried translocation upon NGS. Vingiani et al. [43] analyzed the collection composed of several tumor types. The presence of *NTRK* fusions was confirmed only in 11/30 (37%) specimens identified as candidates by IHC. It is desirable to compare the performance of IHC with our *NTRK* testing procedure in future studies.

Cost considerations are highly important for the comparison of various diagnostic techniques, as they significantly impact the overall treatment budget. This is particularly relevant for rare actionable genetic alterations, given that only a small number of analyzed patients are eventually eligible for targeted therapy [7]. Pricing for laboratory reagents and molecular tests may vary by an order of magnitude between different countries; nevertheless, some rough comparison is possible. The cost of RNA-based NGS per sample is generally above 300–400 EURO or USD [23,44]. IHC is often regarded as a relatively cheap method, although its budget usually exceeds 150 EURO/USD [44]. The price for a single PCR reaction may be around 1.5 EURO/USD or even lower [45]. The *NTRK* testing pipeline presented in this study includes 1 assay for the cDNA quality check, 4 tests for the expression imbalance, and, whenever feasible, 2–8 multiplexed variant-specific PCRs; therefore, the overall expenses for reagents fall within 8–20 EURO/USD. This budget may double in the case of NSCLC analysis, because lung carcinomas have to be additionally tested for *EGFR*, *BRAF*, *KRAS*, *MET*, and *HER2* mutations as well as *ALK*, *ROS1*, and *RET* translocations [17,18,46,47]. Therefore, the described diagnostic approach may be financially advantageous as compared to NGS, even assuming that a small subset of samples still require additional testing by alternative technologies. Within this study, only 21/7424 (0.3%) samples had to undergo further analysis after PCR, and this estimate may become approximately twice as high when considering *ALK*, *ROS1*, and *RET* translocations in addition to *NTRK1*, *NTRK2*, and *NTRK3* rearrangements.

In summary, our report describes a pipeline for the detection of *NTRK1*, *NTRK2*, and *NTRK3* gene fusions. As a screening step, this procedure relies on four PCR tests for 5′/3′-end unbalanced expression (one for *NTRK1*, one for *NTRK2*, and two for *NTRK3*), followed by PCR-driven identification of common translocations. It is advisable to supplement the screening by two multiplexed variant-specific PCRs for *ETV6::NTRK3* and *EML4::NTRK3* rearrangements, respectively. RNA-based NGS is to be applied only to tumors, which demonstrate *NTRK* 5′/3′-end expression imbalances but are negative upon variant-specific PCR. The described approach may be considered to be a cost-efficient alternative to conventional methods of *NTRK* analysis and, therefore, deserves to be evaluated for interlaboratory reproducibility and validity against other methods. For NSCLC, this procedure may be combined with similarly designed PCR-based methods of detecting *ALK*, *ROS1*, *RET*, and *MET* alterations [17,18,47], thus reducing the total cost of molecular testing.

## 4. Materials and Methods

The study included three categories of patients receiving molecular genetic testing in the N.N. Petrov Institute of Oncology between the years of 2019–2022. In particular, we analyzed 6980 consecutive non-small cell lung cancer (NSCLC) cases forwarded for regular testing of actionable genetic events. In addition, *NTRK* testing was applied to 158 pediatric cancer patients; there were 46 sarcomas, 33 inflammatory myofibroblastic tumors, 28 gliomas or neuroepithelial tumors, 4 salivary gland carcinomas, 4 mesoblastic nephromas, 2 thyroid carcinomas, and 41 tumors of other types. The third group was composed of diverse tumor types, which were subjected to testing either due to a known elevated occurrence of *NTRK* translocations or due to the preference of the treating physician (*n* = 937). It included colorectal carcinomas (*n* = 133), sarcomas (*n* = 118), brain tumors (*n* = 104), salivary gland carcinomas (*n* = 102), breast tumors (*n* = 54), thyroid carcinomas (*n* = 30), and other malignancies. Details on sarcoma and salivary gland carcinoma histology are presented in Appendix A.

Formalin-fixed tissue specimens were subjected to manual dissection of tumor cells followed by DNA/RNA isolation and cDNA synthesis, as described previously [17]. Briefly, two–three 10 μm thick tissue sections were lysed for 5–6 h in 200 μL of 10 mM Tris–HCl (pH 8.0), 0.1 mM EDTA (pH 8.0), 2% SDS, and 500 μg/mL proteinase K at 65 °C. The organic extraction was performed with 200 μL Trizol and 90 μL chloroform–isoamyl alcohol mix (24:1). The samples were centrifuged, and the supernatant was incubated with 1 μL of glycogen (20 mg/mL) and 300 μL of isopropanol overnight at −20 °C. Nucleic acids were pelleted by centrifugation, rinsed with 70% ethanol and dissolved in 10 μL of sterile water. cDNA was obtained by the addition of 5× reverse transcriptase reaction buffer, 200 units of RevertAid Reverse Transcriptase (Thermo Fisher Scientific, Waltham, MA, USA), 20 units of RiboCare RNase Inhibitor (Evrogen, Moscow, Russia), dNTP mix (20 nM each), and random hexamers (0.25 μmol). Primer annealing was achieved by incubating the mix at 70 °C, 65 °C, and 60 °C for 5 min, then the reaction was cooled at 0 °C for 2 min. After addition of the enzyme, cDNA synthesis was performed at 20 °C for 5 min and 38 °C for 30 min; the reaction was terminated by heating at 95 °C for 5 min. Quality control relied on PCR amplification of *SDHA* gene fragments; samples producing PCR product after 35 cycles (cycle threshold, Ct) were considered unsuitable for further analysis. Primers and probes for the tests for 5′/3′-end unbalanced expression are given in Appendix A; deltaCt > 3 was taken as threshold for the expression imbalance [16,17,18]. Primers and probes for variant-specific PCR for the 44 most common translocations (involving *TPM3ex7–10* and *NTRK1ex9,10,12*; *BCANex12* and *NTRK1ex10*; *LMNAex3,4,8,10,11* and *NTRK1ex10–12*; *IRF2BP2ex1* and *NTRK1ex8,10*; *BCRex1* and *NTRK2ex17; SQSTM1ex5–6* and *NTRK2ex16*; *ETV6ex4–6* and *NTRK3ex13–15*; *EML4ex2* and *NTRK3ex13,14)*, and the corresponding transcript names, are given in Appendix A. PCR reactions for unbalanced expression of each *NTRK* gene were performed separately. Variant-specific PCR tests were multiplexed in eight reactions according to the involved *NTRK* gene and gene-partner: *LMNA::NTRK1*, *BCAN::NTRK1*, *IRF2BP2::NTRK1*, *TPM3::NTRK1*, *BCR::NTRK2*, *SQSTM1::NTRK2*, *ETV6::NTRK3*, and *EML4::NTRK3*. PCR reactions contained 1 μL of cDNA, 1× GeneAmp PCR Buffer I (Applied Biosystems, Waltham, MA, USA), 250 mkM of each dNTP, 200 nM of each primer and probe, 2.5 mM MgCl_2_, and 1 U of TaqM polymerase (AlkorBio, Saint-Petersburg, Russia) in a total volume of 20 μL. PCR started from enzyme activation (95 °C, 10 min.) and included 38 cycles (95 °C for 15 s followed by 58 °C for 1 min.).

NGS analysis was performed for the samples with 5′/3′-end unbalanced expression in which the type of translocation could not be identified by variant-specific PCR. Tissue samples were subjected to RNA extraction with the PureLink FFPE RNA Isolation kit (Invitrogen, Carlsbad, CA, USA). The analysis was performed with the TruSight RNA Fusion Panel using NextSeq 500 instrument (Illumina, San Diego, CA, USA).

## Figures and Tables

**Figure 1 ijms-24-14203-f001:**
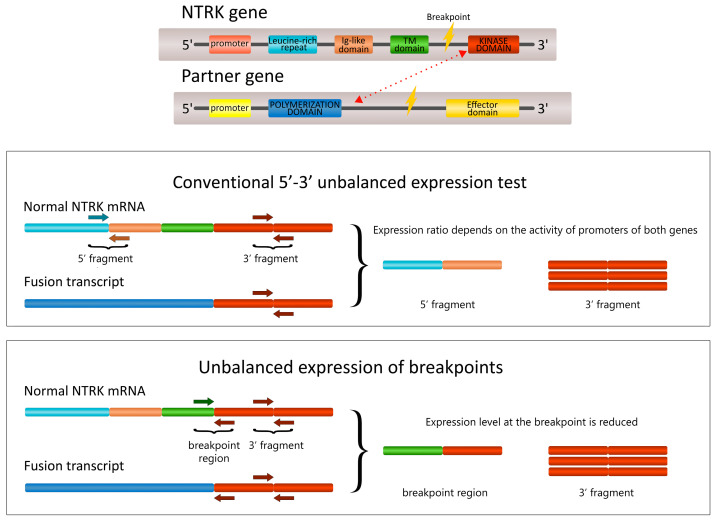
The design of the tests for *NTRK* unbalanced 5′/3′-end expression. The conventional test relies on the overexpression of the kinase portion of the gene upon fusion (**top**); consequently, the 3′-end-specific fragment is overrepresented in the rearranged transcript as compared to the fragment corresponding to the beginning of the gene. The novel version of the test relies on the depletion of expression of the portion of the gene, which is disrupted by the breakpoint and, therefore, absent in the rearranged *NTRK* transcript (**bottom**). Colors represent various portions of the genes. Arrows represent PCR primers.

**Figure 2 ijms-24-14203-f002:**
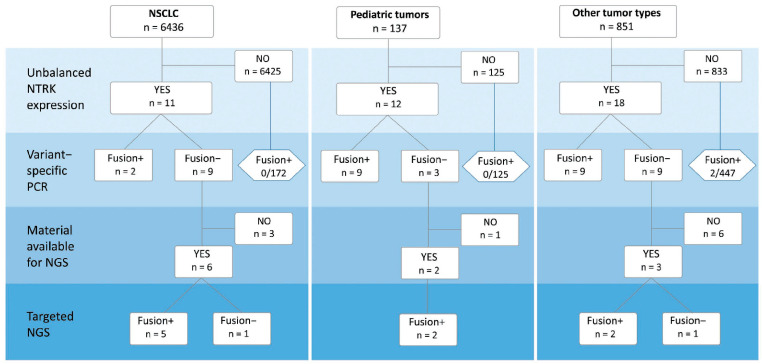
Flow chart of the study.

**Figure 3 ijms-24-14203-f003:**
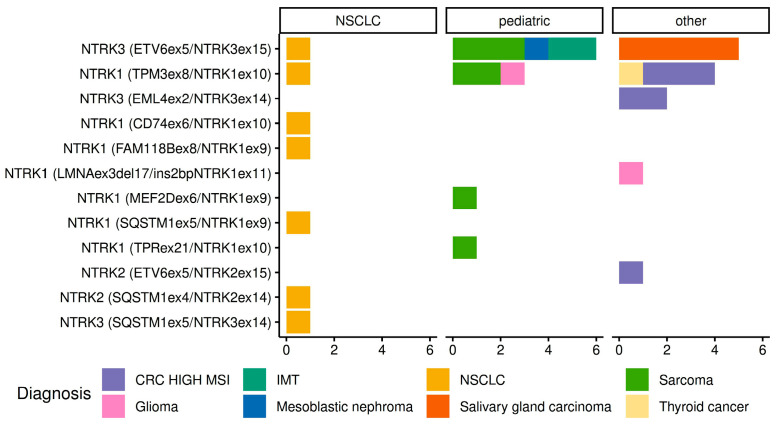
Distribution and spectrum of the identified *NTRK* fusions in different tumor types.

**Table 1 ijms-24-14203-t001:** Frequency of *NTRK* fusions in different tumor types.

Tumor Type	Number of Cases with Confirmed *NTRK* Fusion
NSCLC	7/6436 ^1^ (0.1%)
Sarcoma	7/143 ^2^ (4.9%)
Glioma, neuroepithelial tumor	2/124 (1.6%)
Salivary gland carcinoma	5/93 ^3^ (5.4%)
MSI-high CRC	6/48 (12.5%)
IMT	2/34 (5.9%)
Thyroid cancer	1/26 (3.9%)
Mesoblastic nephroma	1/4 (25.0%)
Other	0/516 (0.0%)

^1^ All fusions were identified in *EGFR/ALK/ROS1/RET/MET*-negative cases (*n* = 5134). ^2^ The studied sarcoma cohort included 3 infantile fibrosarcomas, 5 malignant peripheral nerve sheath tumors, and 2 dermatofibrosarcomas protuberans. ^3^ The studied salivary gland tumor cohort included 2 cases of secretory carcinoma.

**Table 2 ijms-24-14203-t002:** Clinical data of cases with identified *NTRK* fusions.

ID	Tumor Type	Age	Gender	Fusion Type
		*NTRK1*		
22240	CRC, MSI-high	60	f	* TPM3::NTRK1 (T8;N10) *
31589	CRC, MSI-high	64	f	* TPM3::NTRK1 (T8;N10) *
21693	CRC, MSI-high	85	m	* TPM3::NTRK1 (T8;N10) *
28114	Glioblastoma	62	f	* LMNA::NTRK1 (L3del17;ins2N11) *
27845	Oligodendroglioma	3	f	* TPM3::NTRK1 (T8;N10) *
15197	NSCLC	47	f	* CD74::NTRK1 (C6;N10) *
10580	NSCLC	64	m	* FAM118B::NTRK1 (F8;N9) *
12285	NSCLC	64	f	* SQSTM1::NTRK1 (S5;N9) *
14281	NSCLC	60	f	* TPM3::NTRK1 (T8;N10) *
14131	Dermatofibrosarcoma protuberans	6	m	* TPM3::NTRK1 (T8;N10) *
30476	Sarcoma, NOS ^1^	1	f	* TPR::NTRK1 (T21;N10) *
16009	Infantile fibrosarcoma	2	m	* TPM3::NTRK1 (T8;N10) *
13669	Malignant peripheral nerve sheath tumor	0	f	* MEF2D::NTRK1 (M6;N9) *
353067	Papillary thyroid cancer	19	f	* TPM3::NTRK1 (T8;N10) *
		*NTRK2*		
29665	CRC, MSI-high	74	m	* ETV6::NTRK2 (E5;N15) *
13098	NSCLC	79	m	* SQSTM1::NTRK2 (S4;N14) *
		*NTRK3*		
13666	Congenital mesoblastic nephroma	0	m	* ETV6::NTRK3 (E5;N15) *
24631	CRC, MSI-high	68	f	* EML4::NTRK3 (E2;N14) *
33968	CRC, MSI-high	70	f	* EML4::NTRK3 (E2;N14) *
21856	IMT	3	m	* ETV6::NTRK3 (E5;N15) *
12193	IMT	17	m	* ETV6::NTRK3 (E5;N15) *
11479	NSCLC	58	m	* ETV6::NTRK3 (E5;N15) *
22443	NSCLC	44	m	* SQSTM1::NTRK3 (S5;N14) *
32662	Salivary gland adenocarcinoma, NOS	55	f	* ETV6::NTRK3 (E5;N15) *
33039	Salivary gland adenocarcinoma, NOS	32	f	* ETV6::NTRK3 (E5;N15) *
787	Salivary duct carcinoma	65	f	* ETV6::NTRK3 (E5;N15) *
798	Salivary duct carcinoma	49	f	* ETV6::NTRK3 (E5;N15) *
24388	Salivary gland secretory carcinoma	37	f	* ETV6::NTRK3 (E5;N15) *
12471	Soft tissue sarcoma	3	m	* ETV6::NTRK3 (E5;N15) *
16011	Hemangioendothelioma	4	f	* ETV6::NTRK3 (E5;N15) *
16396	Infantile fibrosarcoma	1	m	*ETV6::NTRK3 (E5;N15)*

^1^ NOS—not otherwise specified.

## Data Availability

The data that support the findings of this study are available from the corresponding author upon reasonable request.

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
