# Peer review of "Cost-Efficient Detection of NTRK1/2/3 Gene Fusions: Single-Center Analysis of 8075 Tumor Samples"

_ijms, 2023, doi:10.3390/ijms241814203_

Round 1

Reviewer 1 Report

The authors present an interesting alternative approach for NTRK1-3 rearrangement, by means of unbalanced 5’/3’-end expression analysis by PCR.

The authors state that this alternative approach is cost-effective. However, no cost calculations of the entire workflow have been made. The series of tumor samples analyzed by the approach has a disproportionately high percentage of non-small cell lung carcinoma cases (6436 samples), a tumor types requiring the analysis of rearrangements and splice variants in a substantial number of other genes (ALK, ROS1, RET and METex14 skipping) as well. It doesn’t seem cost-effective to use the proposed test for NTRK1-3, while requiring alternative (RNA seq? additional PCR? FISH?) tests to screen for the other fusion genes as well. In current practice, many laboratories use targeted RNA seq to test for all the fusions/splice variants (also including novel promising targets such as NRG-1 or the FGFR genes). It is highly questionable that the proposed approach is truly cost-effective as compared to RNA-seq in NSCLC.

Although the authors state otherwise, the failure rate of the proposed NTRK 1-3 test is relatively high (the authors report 8.1%), especially as it only measures NTRK. What is the total failure rate if samples failing confirmation are included or even wider, if all molecular markers are to be tested in NSCLC? Using RNA sequencing to test for all relevant gene fusions/splice variants in NSCLC, including NTRK1-3, ALK, ROS1, RET, METex14skip and emerging fusions NRG-1, FGFR1-3, has a failure rate of 5% in our laboratory. It would be important to consider the entire molecular diagnostic workflow for this tumor type when comparisons are made.

The reported numbers of NTRK positivity should be clarified: while the text mentions 31 NTRK fusions, Figure 2 mentions the detection of unbalanced NTRK expression in 11 NSCLC, 12 Pediatric tumors and 18 other tumor types (n = 41). According to Figure 2, confirmation using variant-specific PCR and/or NGS resulted in a final number of 29 confirmed fusions. Does the 31 reflect the 2 “false-negative” cases picked-up by variant-specific PCR in a subset of “expression-negative cases”? This should be clarified in the manuscript. If several techniques are truly required to finally detect NTRK positive cases, it is highly questionable that the proposed approach is truly cost-effective, especially for NSCLC. Moreover, substantial material will be required when multiple tests are needed. In the material and methods section, no information is given on RNA input requirements for the different tests used. Also the time period of case inclusion is not given.

Some sarcoma entities are specifically included in the tables (e.g. MPNST, infantile fibrosarcoma,…) while other cases are generally called “sarcoma”. The same is true for the pediatric sarcomas. Which subtypes were positive? Please specify the latter. Are these considered “Sarcoma NOS”? As for the salivary gland carcinomas, what was the precise histology? Secretory carcinoma? These specifications are important to interpret the results correctly. Please include more accurate histology details on the NTRK positive tumors.

The authors evaluated to what extent the 5’/3’-end NTRK unbalanced expression test is prone to false-negativity only by applying variant-specific PCR for the 44 most common NTRK translocations in a (random?) subset of 744 cases. Although only 2 “false-negative” cases were revealed, this analysis is not really accurate nor representative as no solid comparison to more comprehensive RNA sequencing panels (current “gold-standard”, allowing the detection of unknown fusion partners) were made.  

Author Response

Comment: The authors state that this alternative approach is cost-effective. However, no cost calculations of the entire workflow have been made. The series of tumor samples analyzed by the approach has a disproportionately high percentage of non-small cell lung carcinoma cases (6436 samples), a tumor types requiring the analysis of rearrangements and splice variants in a substantial number of other genes (ALK, ROS1, RET and METex14 skipping) as well. It doesn’t seem cost-effective to use the proposed test for NTRK1-3, while requiring alternative (RNA seq? additional PCR? FISH?) tests to screen for the other fusion genes as well. In current practice, many laboratories use targeted RNA seq to test for all the fusions/splice variants (also including novel promising targets such as NRG-1 or the FGFR genes). It is highly questionable that the proposed approach is truly cost-effective as compared to RNA-seq in NSCLC.

Response: We have now incorporated the paragraph on the cost calculations in the Discussion:

Cost considerations are highly important for the comparison of various diagnostic techniques, as they significantly impact the overall treatment budget. This is particularly relevant for rare actionable genetic alterations, given that only a small number of analyzed patients are eventually eligible for targeted therapy [7]. Pricing for laboratory reagents and molecular tests may vary by an order of magnitude between different countries, nevertheless some rough comparison is possible. The cost of RNA-based NGS per sample is generally above 300-400 EURO or USD [23,42]. The price for a single PCR reaction may be around 1.5 EURO/USD or even lower [43]. The NTRK testing pipe-line presented in this study includes 1 assay for the cDNA quality check, 4 tests for the expression imbalance, and, whenever feasible, 2-8 multiplexed variant-specific PCRs, therefore the overall expenses for reagents fall within 8-20 EURO/USD. This budget may double in case of NSCLC analysis, because lung carcinomas have to be additionally tested for EGFR, BRAF, KRAS, MET and HER2 mutations as well as ALK, ROS1 and RET translocations [17,18,44,45]. Therefore, the described diagnostic approach may be financially advantageous as compared to NGS, even assuming that a small subset of samples still require additional testing by alternative technologies. Within this study, only 21/7424 (0.3%) samples had to undergo  further analysis after PCR, and this estimate may become approximately twice higher when considering ALK, ROS1 and RET translocations in addition to NTRK1, NTRK2, and NTRK3 rearrangements.

For NSCLC, this procedure may be combined with similarly designed PCR-based methods of detecting ALK, ROS1, RET, and MET alterations [17,18,45], thus reducing the total cost of molecular testing.

Comment: Although the authors state otherwise, the failure rate of the proposed NTRK 1-3 test is relatively high (the authors report 8.1%), especially as it only measures NTRK. What is the total failure rate if samples failing confirmation are included or even wider, if all molecular markers are to be tested in NSCLC? Using RNA sequencing to test for all relevant gene fusions/splice variants in NSCLC, including NTRK1-3, ALK, ROS1, RET, METex14skip and emerging fusions NRG-1, FGFR1-3, has a failure rate of 5% in our laboratory. It would be important to consider the entire molecular diagnostic workflow for this tumor type when comparisons are made.

Response: We have now incorporated appropriate comments in the Discussion:

… Importantly, 91.9% of the included tumors passed the RNA quality control for this test. These data are comparable with the estimates obtained in the NGS RNA studies [19-23]. The most common cause of the RNA isolation failure is poor processing of morphological samples, which is still an issue in many hospitals [24]. It is highly likely that RNA/cDNA samples, which are not suitable for reliable PCR-based testing, will also not be compatible with RNA-based NGS analysis. This study considered biological samples arriving from a few dozen different cancer centers, therefore some variations in the tissue handling could have inflated the failure rate.

Comment: The reported numbers of NTRK positivity should be clarified: while the text mentions 31 NTRK fusions, Figure 2 mentions the detection of unbalanced NTRK expression in 11 NSCLC, 12 Pediatric tumors and 18 other tumor types (n = 41). According to Figure 2, confirmation using variant-specific PCR and/or NGS resulted in a final number of 29 confirmed fusions. Does the 31 reflect the 2 “false-negative” cases picked-up by variant-specific PCR in a subset of “expression-negative cases”? This should be clarified in the manuscript.

Response: Figure 2, indeed, described the workflow resulting in the identification of 29 fusions and did not show 2 “false-negative” cases. In order to eliminate this and other uncertainties, this Figure has been modified, so now it describes the entire workflow.

Comment: If several techniques are truly required to finally detect NTRK positive cases, it is highly questionable that the proposed approach is truly cost-effective, especially for NSCLC. Moreover, substantial material will be required when multiple tests are needed.

Response: Please consider our response to the Comment #1 (see above).

Comment: In the material and methods section, no information is given on RNA input requirements for the different tests used. Also the time period of case inclusion is not given.

Response: We now mention in the Discussion the time period for case inclusion. We have also specified all relevant details concerning RNA isolation, cDNA synthesis and composition of PCR tests:

Briefly, two-three 10 μm-thick tissue sections were lysed for 5-6 hours in 200 μl of 10 mM Tris–HCl (pH 8.0), 0.1 mM EDTA (pH 8.0), 2% SDS and 500 μg/ml proteinase K at 65°C. The organic extraction was performed with 200 μL Trizol and 90 μL chloroform–isoamyl alcohol mix (24:1). The samples were centrifuged, and the supernatant was incubated with 1 μL of glycogen (20 mg/mL) and 300 μL of isopropanol overnight at −20°C. Nucleic acids were pelleted by centrifugation, rinsed with 70% ethanol and dissolved in 10 μL of sterile water. cDNA was obtained by the addition of 5X reverse transcriptase reaction buffer, 200 units of RevertAid Reverse Transcriptase (Thermo Fisher Scientific, USA), 20 units of RiboCare RNase Inhibitor (Evrogen, Russia), dNTP mix (20nM each), random hexamers (0.25 μmol). Primer annealing was achieved by incubating the mix for 5 min. at 70°Ð¡, 65°Ð¡ and 60°Ð¡ for 5 min., then the reaction was cooled at 0˚С for 2 min. After addition of the enzyme, cDNA synthesis was performed at 20°Ð¡ for 5 min and 38°Ð¡ for 30 min; the reaction was terminated by heating at 95°Ð¡ for 5 min. ….. PCR reactions contained 1 μl of cDNA, 1× GeneAmp PCR Buffer I (Applied Biosystems, USA), 250 mkM of each dNTP, 200 nM of each primer and probe, 2.5 mM MgCl2, and 1 U of TaqM polymerase (AlkorBio, Russia) in a total volume of 20 μl. PCR started from enzyme activation (95°Ð¡, 10 min.), and included 38 cycles (95°Ð¡ for 15 s followed by 58°Ð¡ for 1 min.).

Comment: Some sarcoma entities are specifically included in the tables (e.g. MPNST, infantile fibrosarcoma,…) while other cases are generally called “sarcoma”. The same is true for the pediatric sarcomas. Which subtypes were positive? Please specify the latter. Are these considered “Sarcoma NOS”? As for the salivary gland carcinomas, what was the precise histology? Secretory carcinoma? These specifications are important to interpret the results correctly. Please include more accurate histology details on the NTRK positive tumors.

Response: We have now inserted these specifications in Table 1, Table 2 and Table S1. We also now mention in the text that “… fusions were detected in 2/3 infantile fibrosarcomas, 1/5 malignant peripheral nerve sheath tumors, 1/2 dermatofibrosarcomas protuberans, 1/1 hemangioendothelioma, and 2/8 sarcomas not otherwise specified.”

Comment: The authors evaluated to what extent the 5’/3’-end NTRK unbalanced expression test is prone to false-negativity only by applying variant-specific PCR for the 44 most common NTRK translocations in a (random?) subset of 744 cases. Although only 2 “false-negative” cases were revealed, this analysis is not really accurate nor representative as no solid comparison to more comprehensive RNA sequencing panels (current “gold-standard”, allowing the detection of unknown fusion partners) were made.

Response: We have now incorporated appropriate comments in the Discussion:

This investigation did not involve thorough validation of NTRK–rearranged and NTRK-wild-type samples by RNA sequencing, which is a limitation of the study. While the confirmation of NTRK fusions detected by the combination of the 5’/’3-end expression imbalance assay and variant-specific PCR seems unnecessary, it is unclear what is the risk of missing some rare NTRK rearrangements by the described above pipe-line. The frequency of NTRK fusions is low, therefore, it is challenging to ensure that any given technology does not produce false-negative results. This validation would require RNA-based NGS analysis involving huge tumor collections, and therefore, excessive costs. It is noteworthy that the frequencies of NTRK rearrangements presented in this data set match the previously reported NTRK prevalence estimates, thus supporting the suitability of the 5’-3’ NTRK imbalance assay [25,28,35-37,39,40]. Furthermore, we have encouraging results of the NSCLC study, which focused on patients with a very high probability of the presence of actionable gene fusion in the tumor tissue, i.e., young-onset female non-smokers. RNA-based NGS was performed for 87 tumors, which lacked activating mutations in EGFR, KRAS, NRAS, BRAF, MET and HER2 oncogenes and were negative for ALK, ROS1, RET or NTRK1/2/3 translocations by the 5’/3’-end expression test and variant-specific PCR. Strikingly, no instances of rearrangements in the above genes have been revealed by RNA sequencing [41]. While data on NSCLC appear to be conclusive, we did not specifically address the performance of the NTRK unbalanced expression assay separately in other tumor types. There are significant differences in the patterns of NTRK expression and spectrum of NTRK rearrangements across various categories of malignancies [3-6,8-10], which may influence the performance of this test. This is particularly relevant to tumor entities, which have high base-line expression level of NTRK kinases [26], as the presence of the background normal transcripts is likely to compromise the performance of the 5’/3’-end expression assay. 

Reviewer 2 Report

The authors propose a relatively simple and cheap approach to identification of NTRK fusions in a variety of clinical specimens. This is an important topic, given how expensive it is to provide RNA NGS-based testing for all tumours in the clinical setting. However, it is critical that a clinical assay is completely and properly validated prior to use, and there are some important aspects that have not been addressed in the current manuscript.

1.     The analysis of the imbalance assay for identifying fusion genes has been demonstrated through other assays to be highly gene specific.  In other words, the cut off for the ratio that designates “over-expressed” can be very different for each gene.  The paper did not show any evidence that the authors established such ratios for the three assays in question- in the materials and methods, there is a statement that deltaCt>3 was considered imbalanced, but no information provided about how that number was chosen and why it is the same for each of the 3 genes. The authors should explain their method of validating the assay. This all needs to be included in the publication.

2.     The design of the unbalanced expression of breakpoints assay is inherently flawed if there is wildtype RNA in the assay, which there almost always is. While the authors note this in the discussion, they do not provide a mechanism to avoid false negatives that would stem from this.  As a clinical assay, this level of uncertainty would be unacceptable.

3.     Of 6436 NSCLC samples, 11 showed ratios suggesting the presence of a fusion. Of these, 7 were confirmed by another method, 1 was shown to be negative by another method and 3 could not be confirmed. This translates to 87.5% of evaluable cases confirmed as positives. What sensitivity is expected for a clinical assay?  Is this good enough?

4.     To check the false negative rate, the authors evaluated 172 NSCLC, 125 pediatric tumours and 447 other tumours.  From this set of 744, 2 positives were identified – but this was done using only the variant-specific analysis approach, which the authors had just demonstrated to be not highly sensitive.  (From Figure 2, the variant specific approach identified 2 of 11, 9 of 12 and 9 of 18 positives from these three categories when evaluating the positives). Given the relatively poor performance of the variant specific assay, it would seem this is not a good test to check for false negatives. RNA-based NGS should have been used, or at the very least, IHC.

5.     It is not possible to provide the false negative rate without looking at many more samples.  If the authors had done a more conventional validation, where they used their imbalance assay on a large set of samples for which the fusion status was already defined, there could be more confidence in the results and it would be possible to  evaluate sensitivity, specificity, reproducibility, etc.  Without that, it is not possible to know how well these assays are performing. Certainly, calculating the proportion of the ~8000 tumours that are positive NTRK fusions without checking ALL of the negative cases by another method is not supported.

6.     Table 1 is confusing.  The text says they look at 6436 NSCLC samples, and identified 11 unbalanced assays.  The table says they looked at 6436 and identified 7 unbalanced assays. That table also shows numbers that are not in the text about the negative cases tested,  but does not indicate which ones came up positive.

7.     The figures are very hard to read- the type is very small and not in focus.

8.     There are grammatical errors that should be corrected prior to publication.

Some grammatical editing is required.

Author Response

Comment: The analysis of the imbalance assay for identifying fusion genes has been demonstrated through other assays to be highly gene specific.  In other words, the cut off for the ratio that designates “over-expressed” can be very different for each gene.  The paper did not show any evidence that the authors established such ratios for the three assays in question- in the materials and methods, there is a statement that deltaCt>3 was considered imbalanced, but no information provided about how that number was chosen and why it is the same for each of the 3 genes. The authors should explain their method of validating the assay. This all needs to be included in the publication.

Response: We have performed some additional experiments, which are now described in the Results section, and have added Figure S1 to the article:

The performance of the 5’/3’-end NTRK unbalanced expression assay depends on the chosen threshold. We utilized deltaCt > 3 as a cut-off, given that this approach produced satisfactory results for the testing of other rearranged kinases [16,18,19]. We evaluated the reliability of this threshold by comparing samples with known NTRK rearrangements against tumors with presumably normal NTRK status (Figure S1). We were able to obtain in our tumor bank 39 samples with NTRK1 or NTRK3 fusions in which the presence of alteration was confirmed by variant-specific PCR or NGS. However, we could not perform the validation study for NTRK2, as only two samples with NTRK2 fusions were available. The control group was composed of 50 NSCLCs, which carried alterations in EGFR, ALK, RET, ROS1, MET oncogenes and, therefore, were highly unlikely to have NTRK rearrangements. For NTRK1, 10/10 (100%) NTRK1 fusion-containing samples were correctly identified by the unbalanced NTRK1 test (Figure S1A). The test evaluating the depletion of NTRK3 ex13-14 junction detected only 2/7 (28.6%) tumors with NTRK3 fusions involving this breakpoint (Figure S1B). The test for NTRK3 ex14-15 depletion identified 15/22 (68.2%) cases with translocations involving the breakpoint at exon 15 (Figure S1C). As the chosen threshold (deltaCt > 3) performed poorly for both NTRK3 assays, we attempted to evaluate whether the modification of the cut-off will improve the testing procedure. We performed PCR analysis for 11 individual NTRK3 translocation variants in 225 cases showing either deltaCt > 0.9 for NTRK3 ex13-14 depletion assay or deltaCt > 1.5 for NTRK3 ex14-15 depletion assay, however none of these samples with low-level NTRK3 expression imbalance carried NTRK3 translocations.

Comment: The design of the unbalanced expression of breakpoints assay is inherently flawed if there is wildtype RNA in the assay, which there almost always is. While the authors note this in the discussion, they do not provide a mechanism to avoid false negatives that would stem from this.  As a clinical assay, this level of uncertainty would be unacceptable.

Response: The mentioned issue is now more clearly stated in the Discussion as one of the limitations of the study:

This investigation did not involve thorough validation of NTRK–rearranged and NTRK-wild-type samples by RNA sequencing, which is a limitation of the study. While the confirmation of NTRK fusions detected by the combination of the 5’/’3-end expression imbalance assay and variant-specific PCR seems unnecessary, it is unclear what is the risk of missing some rare NTRK rearrangements by the described above pipe-line. The frequency of NTRK fusions is low, therefore, it is challenging to ensure that any given technology does not produce false-negative results. This validation would require RNA-based NGS analysis involving huge tumor collections, and therefore, excessive costs. It is noteworthy that the frequencies of NTRK rearrangements presented in this data set match the previously reported NTRK prevalence estimates, thus supporting the suitability of the 5’-3’ NTRK imbalance assay [25,28,35-37,39,40]. Furthermore, we have encouraging results of the NSCLC study, which focused on patients with a very high probability of the presence of actionable gene fusion in the tumor tissue, i.e., young-onset female non-smokers. RNA-based NGS was performed for 87 tumors, which lacked activating mutations in EGFR, KRAS, NRAS, BRAF, MET and HER2 oncogenes and were negative for ALK, ROS1, RET or NTRK1/2/3 translocations by the 5’/3’-end expression test and variant-specific PCR. Strikingly, no instances of rearrangements in the above genes have been revealed by RNA sequencing [41]. While data on NSCLC appear to be conclusive, we did not specifically address the performance of the NTRK unbalanced expression assay separately in other tumor types. There are significant differences in the patterns of NTRK expression and spectrum of NTRK rearrangements across various categories of malignancies [3-6,8-10], which may influence the performance of this test. This is particularly relevant to tumor entities, which have high base-line expression level of NTRK kinases [26], as the presence of the background normal transcripts is likely to compromise the performance of the 5’/3’-end expression assay. 

Comment: Of 6436 NSCLC samples, 11 showed ratios suggesting the presence of a fusion. Of these, 7 were confirmed by another method, 1 was shown to be negative by another method and 3 could not be confirmed. This translates to 87.5% of evaluable cases confirmed as positives. What sensitivity is expected for a clinical assay?  Is this good enough?

Response: Overall, we believe that false-positives are not a major issue in the described protocol, as the overall number of samples demonstrating the 5’/3’-end unbalanced expression is relatively low. We comment on this issue on the Discussion:

There were two tumors, which demonstrated clear-cut 5’/3’-end NTRK2 (NSCLC) and NTRK3 (adult sarcoma) expression imbalance, but turned out to be NTRK rearrangement-negative by the TruSight RNA Fusion NGS panel. It is unclear if these tumors utilize alternative splicing for one of the NTRK genes, or if these observations are related to the NGS failures in identifying NTRK fusions. It is noteworthy that NGS testing revealed a potentially actionable EWSR1::CCDC80 fusion in the described above sarcoma, thus making unlikely the existence of NTRK3 translocation.

Comment: To check the false negative rate, the authors evaluated 172 NSCLC, 125 pediatric tumours and 447 other tumours.  From this set of 744, 2 positives were identified – but this was done using only the variant-specific analysis approach, which the authors had just demonstrated to be not highly sensitive.  (From Figure 2, the variant specific approach identified 2 of 11, 9 of 12 and 9 of 18 positives from these three categories when evaluating the positives). Given the relatively poor performance of the variant specific assay, it would seem this is not a good test to check for false negatives. RNA-based NGS should have been used, or at the very least, IHC.

Response: Please see our response to the Comment #2, which addresses this issue.

Comment: It is not possible to provide the false negative rate without looking at many more samples.  If the authors had done a more conventional validation, where they used their imbalance assay on a large set of samples for which the fusion status was already defined, there could be more confidence in the results and it would be possible to evaluate sensitivity, specificity, reproducibility, etc. Without that, it is not possible to know how well these assays are performing. Certainly, calculating the proportion of the ~8000 tumours that are positive NTRK fusions without checking ALL of the negative cases by another method is not supported.

Response: We agree with this criticism, we have now addressed this issue in the Discussion  (Please see our response to the Comment #2). In addition, we have added the following statement: The described approach … deserves to be evaluated for inter-laboratory reproducibility and validity against other methods.

Comment: Table 1 is confusing.  The text says they look at 6436 NSCLC samples, and identified 11 unbalanced assays.  The table says they looked at 6436 and identified 7 unbalanced assays. That table also shows numbers that are not in the text about the negative cases tested,  but does not indicate which ones came up positive.

Response: We have removed the 2nd column to confusion and added some additional clarifications.

Comment: The figures are very hard to read- the type is very small and not in focus.

Response: We have increased the size and the resolution of the Figures. We have also submitted original files, which can be used during the processing of the manuscript.

Comment: There are grammatical errors that should be corrected prior to publication.

Response: The paper has been checked by Dr. Priscilla S. Amankwah, who is a native English speaker proficient in biomedical writing.

Round 2

Reviewer 2 Report

The additional experiments performed to determine the appropriate cut off to identify imbalances for each of the NTRK genes are appreciated.  It is unfortunate that NTRK2 could not be evaluated in this study, and the discussion should include a note that this assay has not been validated for detection of NTRK2 variants.  Right now, the discussion mentions that the authors have designed a “robust” assay for detection of fusions with all 3 genes- I’m not sure this qualifies as robust.  From the description provided, it is not clear to me how well the assay works for NTRK3.  The experiments described suggest there is more work to be done.  If delta cT>3 does not work well for either of the NTRK3 common breakpoints, what does?  If delta cT>3 doesn’t work well for NTRK3, did this change any of the analysis in the manuscript, which was originally presented with that value as the cut off for positivity?

I appreciate that doing full RNA-based NGS is expensive so the true false negative rate is not possible to determine.  This should be clearly stated.  However, it would be possible to do IHC rather cheaply, which could at least help to identify cases through another screening mechanism that could be positive.  If you are going to design an assay that you are saying is a good and robust tool for detecting an analyte, then you need to do the work to show that this is true.  If you can’t do it because it’s too expensive or difficult, then don’t claim to have a robust test.

While I appreciate that this assay, coupled with PCR assays, can pick up some of the NTRK fusions in clinical specimens, and may be suitable for labs with limited budgets,  I am unconvinced that this is a more effective screen than IHC, at least in the setting of non-CNS tumours.   The authors claim in their introduction that many studies have questioned the suitability of IHC for this role- they cite only one study.  In that study, the false negative rate of IHC is given as 18%.  It is not clear how this approach compares to that.  This should perhaps be acknowledged in the discussion.

Author Response

We appreciate greatly your comments, we find them very appropriate. Please see below our responses:

Comment: It is unfortunate that NTRK2 could not be evaluated in this study, and the discussion should include a note that this assay has not been validated for detection of NTRK2 variants.

Response: We have inserted the following statement: “Furthermore, our protocol has not been rigorously validated for the detection of NTRK2 fusions due to low number of “positive” samples, which is another limitation of this study.”  

Comment: Right now, the discussion mentions that the authors have designed a “robust” assay for detection of fusions with all 3 genes- I’m not sure this qualifies as robust. 

Response: We have deleted the word “robust” from the description of our assay.

Comment: From the description provided, it is not clear to me how well the assay works for NTRK3.  The experiments described suggest there is more work to be done.  If delta cT>3 does not work well for either of the NTRK3 common breakpoints, what does?  If delta cT>3 doesn’t work well for NTRK3, did this change any of the analysis in the manuscript, which was originally presented with that value as the cut off for positivity?

Response: We have now provided an additional comment on this issue: “We attempted to decrease the cut-off for deltaCt in order to reduce the number of false-negative results obtained upon the quantitation of the 5’- and 3’-specific NTRK3 transcripts, however, our experiments revealed that the modification of the threshold does not improve the performance of the assay (Supplementary Figure S1).”.

Comment: I appreciate that doing full RNA-based NGS is expensive so the true false negative rate is not possible to determine.  This should be clearly stated. 

Response: We now state that “The actual rate false-negative results produced by our methodology remains to be determined in further investigations”.

Comment: However, it would be possible to do IHC rather cheaply, which could at least help to identify cases through another screening mechanism that could be positive.  If you are going to design an assay that you are saying is a good and robust tool for detecting an analyte, then you need to do the work to show that this is true.  If you can’t do it because it’s too expensive or difficult, then don’t claim to have a robust test.

Response: We have deleted the word “robust” from the description of our test. We also have inserted a comment on the cost of pan-TRK IHC: “IHC is often regarded as a relatively cheap method, although its budget usually exceeds 150 EURO/USD [45]”.

Comment: While I appreciate that this assay, coupled with PCR assays, can pick up some of the NTRK fusions in clinical specimens, and may be suitable for labs with limited budgets,  I am unconvinced that this is a more effective screen than IHC, at least in the setting of non-CNS tumours.   The authors claim in their introduction that many studies have questioned the suitability of IHC for this role- they cite only one study.  In that study, the false negative rate of IHC is given as 18%.  It is not clear how this approach compares to that.  This should perhaps be acknowledged in the discussion.

Response: We have incorporated a paragraph, which describes the performance of IHC pan-TRK testing:  

Many laboratories currently utilize pan-TRK IHC as a screening tool for the detection of NTRK rearrangements [9]. Several investigations evaluated the performance of IHC testing in the real-world setting. Hondelink et al. [8] evaluated the sensitivity of the IHC by analyzing the tumors with NGS-detected NTRK translocations. These authors considered their own data set (24 tumors) and all relevant published studies (200 tumors in total), and revealed that IHC failed to detect fusions in 40/224 (18%) of NTRK-rearranged samples (NTRK1: 6%; NTRK2: 14%; NTRK3: 27%) [8]. These data correspond well to our experience, as we observed satisfactory performance of the 5’-/3’-end expression imbalance screening test for the NTRK1 but not for the NTRK3 (Supplementary Figure 1S). IHC often produces false-positive results. Overbeck et al. [42] reported the results of the pan-TRK staining for 973 NSCLCs. IHC failures were observed for 75/973 (8%) tumors. TRK expression was detected in 133/898 (15%) NSCLCs; 120 of these carcinomas were available for NGS analysis, with only two instances of NTRK fusions eventually confirmed. Zito Marino et al. [43] revealed pan-TRK IHC positivity in 16 out of 83 triple-negative breast malignancies, however none of these samples carried translocation upon NGS. Vingiani et al. [44] analyzed the collection composed of several tumor types; presence of NTRK fusions was confirmed only in 11/30 (37%) specimens identified as candidates by IHC. It is desirable to compare the performance of IHC with our NTRK testing procedure in the future studies.